Patterns of pre-sleep music use and sleep quality: exploratory survey findings on state anxiety

http://orcid.org/0009-0008-6461-7831 Danso Andrew 1 2 andrew.a.dansoadu@jyu.fi
http://orcid.org/0000-0002-8772-8058 Ehlert Mareike 3
http://orcid.org/0000-0002-0877-7100 Koehler Friederike 1 2
http://orcid.org/0000-0001-5499-0271 Kirk Rory 4
Natarajan Nandhini 1 2
http://orcid.org/0000-0001-8159-7467 Wright Shannon Eilyce 5
Timmers Renee 4 6
http://orcid.org/0000-0002-4647-8048 Saarikallio Suvi 1 2
1 Centre of Excellence in Music, Mind, Body and Brain, University of Jyväskylä , Jyväskylä, Central Finland , Finland
2 Department of Music, Art and Culture Studies, University of Jyväskylä , Jyväskylä, Central Finland , Finland
3 Institute for Psychology in Education, University of Münster , Münster, North Rhine-Westphalia , Germany
4 Muses Mind Machine, Department of Music, University of Sheffield , Sheffield, South Yorkshire , United Kingdom
5 Department of Psychology, University of the Fraser Valley , Abbotsford, British Columbia , Canada
6 Department of Music & Healthy Lifespan Institute, University of Sheffield , Sheffield, South Yorkshire , United Kingdom
Farhan Faiza
Electronic publication date: 2025 Nov 26
Publication date: 2025
Volume: 13
Electronic Location ID: e20444
Received 2025 May 5; Accepted 2025 Oct 31
Copyright: © 2025 Danso et al.
Copyright year: 2025
Copyright holder: Danso et al.
License: This is an open access article distributed under the terms of the Creative Commons Attribution License, which permits unrestricted use, distribution, reproduction and adaptation in any medium and for any purpose provided that it is properly attributed. For attribution, the original author(s), title, publication source (PeerJ) and either DOI or URL of the article must be cited.
License URL: https://creativecommons.org/licenses/by/4.0/

Keywords: Sleep quality, Music, Psychological distress, Hyperarousal, Coping, Physical activity

Funding: Research Council of Finland 346210 This project has received funding from the Research Council of Finland [346210]. The funders had no role in study design, data collection and analysis, decision to publish, or preparation of the manuscript.

==============================
Music listening is a widely used self-help approach that may influence psychological and physiological processes associated with sleep. This cross-sectional study explored patterns of pre-sleep music use in relation to psychological distress (state anxiety, mood disturbance, stress) and subjective sleep quality. Adults (N = 269, 52.6% female; Mage = 27.7, SD = 9.0) completed validated self-report measures of sleep quality (the Pittsburgh Sleep Quality Index (PSQI)) and psychological distress. Pre-sleep music use was modestly associated with poorer sleep quality (r = 0.23, p < 0.01). A borderline interaction between state anxiety and music use (β = −0.170, p = 0.050) suggested, but did not confirm, a possible buffering pattern in which the anxiety-sleep association appeared weaker among more frequent music users. No moderation effects were observed for mood or stress. These preliminary findings suggest that pre-sleep music use may reflect a coping-oriented effort among individuals experiencing anxiety. However, given the cross-sectional design, self-report measures, and borderline statistical support, the results should be viewed as descriptive and hypothesis-generating.

Introduction

Sleep problems are frequently managed through self-help approaches, particularly in contexts where formal treatments are unavailable, inaccessible, or deemed undesirable (Ancoli-Israel & Roth, 1999; Morin et al., 2006). Such approaches are considered coping efforts that individuals adopt to regulate sleep-related distress in the absence of, or as an alternative to, formal interventions (Jespersen et al., 2022; Buus, Genovese & Jespersen, 2025). These include reading, relaxation techniques, or listening to music, and are often relied upon for extended periods of time prior to individuals seeking professional help (Morin et al., 2006; Brown, Qin & Esmail, 2017; Jespersen et al., 2022). Perhaps disconcertingly, evidence by Morin et al. (2006) and Buus, Genovese & Jespersen (2025) indicate these behaviors being increasingly common among young people and emerging in response to heightened sleep problems.

Use of music as a self-help approach for managing sleep difficulties

Listening to music is a widely reported self-help approach for managing sleep difficulties (Bjorvatn, Waage & Saxvig, 2023; Buus, Genovese & Jespersen, 2025). Population-based surveys indicate that roughly one quarter of adults use music at least occasionally to aid sleep, though regular use (weekly or daily) is less prevalent (Morin et al., 2006; Buus, Genovese & Jespersen, 2025). Patterns of use appear to be influenced by both age and symptom severity, as younger individuals are significantly more likely to report using music prior to sleep (Buus, Genovese & Jespersen, 2025), and in Norway, music and similar ‘tricks’ were more frequently used by individuals with chronic insomnia than by good sleepers (i.e., individuals without insomnia) (Bjorvatn, Waage & Saxvig, 2023). In student populations, frequent music use has been linked to shorter sleep duration and greater sleep difficulties (Brown, Qin & Esmail, 2017), suggesting it may often emerge as a coping-oriented response to heightened sleep problems. Trahan et al. (2018) reported a high prevalence of music use for sleep, with 62% of respondents indicating they had used music as a sleep aid at least once. Yet, only about one-third of users reported weekly use, and fewer than 15% used music nearly every day, indicating that habitual integration into pre-sleep routines is relatively uncommon.

The effects of music on sleep

Available systematic reviews and meta-analyses suggests that music can function as a transition aid supporting sleep onset, with consistent benefits on subjective outcomes, though evidence from objective measures is less conclusive (De Niet et al., 2009; Wang, Sun & Zang, 2014; De Witte et al., 2020; Yamasato et al., 2020; Jespersen et al., 2022). Sedative music, typically characterized by a tempo of 60–80 beats per minute, low volume, instrumental timbre, and a stable, repetitive structure, has most frequently been associated with relaxation and sleep facilitation (Chang et al., 2012; Chen et al., 2014; Trahan et al., 2018; Lund, 2021; Scarratt et al., 2023).

Data indicate that sedative music may decrease sympathetic arousal pathways, reducing cortisol levels, influencing heart rate and blood pressure, leading to a state conducive to sleep (Harmat, Takács & Bódizs, 2008; Chang et al., 2012; Liu et al., 2016; Jespersen et al., 2022; Zhao, Lund & Jespersen, 2024). In addition, music exerts multiple psychological effects relevant to sleep, including reducing anxiety and stress, improving relaxation, regulating mood, and providing distraction from intrusive thoughts (Harvey, 2002; Buus, Genovese & Jespersen, 2025). Evidence also points to roles in habit formation, comfort, and expectancy effects, which may further support sleep quality (Yamasato et al., 2020; Lund, 2021; Jespersen et al., 2022; Dickson & Schubert, 2022). The aforementioned evidence suggests that music may aid sleep both by lowering physiological arousal and psychological regulation. While structured music interventions (e.g., music-assisted relaxation, which is a relaxation-improving intervention, see De Niet et al., 2009) have demonstrated generalizable benefits for sleep quality (Iwaki, Tanaka & Hori, 2003; De Niet et al., 2009), the effects of self-selected music appear increasingly nuanced, subject to individual preference, familiarity, context of use, and the listener’s intention to sleep (Tan et al., 2012). Evidence indicates that personal preference and familiarity play a strong role in influencing sleep outcomes. For instance, Tan et al. (2012) found that both preference and familiarity were positively associated with perceived relaxation, indicating that music judged as more liked or more familiar was experienced as more relaxing. Iwaki, Tanaka & Hori (2003) reported that preferred familiar music facilitated sleep under natural conditions, yet interfered when sleep was attempted under pressure, consistent with ironic process theory (e.g., monitoring whether you’re falling asleep, see e.g., Wegner, 1994). Trahan et al. (2018) extended these findings into everyday use, reporting that self-selected music for sleep indicate highly individual preferences (rather than shared structural features) and was perceived to aid sleep via pathways such as relaxation, distraction, mood regulation, and habit formation. Critically, these findings suggest that music’s efficacy as a sleep aid depends on personal preference and familiarity, indicating that self-selected music may leverage similar mechanisms as structured music interventions to engage relaxation and sleep (Tan et al., 2012; De Niet et al., 2009).

Psychological distress and sleep difficulties

Psychological distress is broadly defined as a state of emotional suffering or poor health, characterized by symptoms of anxiety, stress, and low mood (Blackwelder, Hoskins & Huber, 2021; Zhang et al., 2022). Psychological distress is consistently linked to poor sleep outcomes, including short sleep duration and reduced subjective sleep quality (Brown, Qin & Esmail, 2017). Psychological distress contributes to impaired functioning as well as physical disability, reduced quality of life, and greater healthcare utilization (Zhao, Lund & Jespersen, 2024). Evidence highlights a bidirectional relationship between psychological distress and sleep difficulties, whereby heightened distress increases the likelihood of sleep disturbances, while poor sleep further exacerbates psychological distress (Blackwelder, Hoskins & Huber, 2021). Epidemiological findings support this relationship across populations (Alimoradi et al., 2021). For instance, during the COVID-19 pandemic, pooled prevalence estimates indicated that 37% of adults reported sleep problems, with particularly high rates among patients with COVID-19 (55%) and healthcare professionals (43%). Among students, symptoms of depression and anxiety have been strongly associated with poor sleep quality and reduced life satisfaction, with around half reporting poor sleep (Seun-Fadipe & Mosaku, 2017).

These findings are consistent with the hyperarousal model, which posits that chronic sleep difficulties arise from conditioned arousal (e.g., anxiety related to bodily sensations when trying to sleep), maladaptive coping behaviors (e.g., excessive time in bed), and heightened cognitive-emotional engagement with sleep problems (e.g., catastrophic thinking about sleep loss), even in non-clinical populations (Bonnet & Arand, 2010; Riemann et al., 2010). In this model, individuals with sleep difficulties exhibit an overactive arousal system—marked by psychological distress and high mental alertness—that interferes with the initiation and maintenance of sleep (Bonnet & Arand, 2010; Riemann et al., 2010). This framework is particularly relevant to subclinical sleep difficulties, where stress, anxiety, and heightened pre-sleep arousal interfere with sleep onset (i.e., falling asleep) and sleep maintenance. Within this context, music listening may serve as a self-help approach to temporarily reduce physiological and psychological tension before sleep.

Coping efforts, behavior patterns and sleep disturbances

Thus, in response to sleep difficulties occurring alongside heightened psychological distress, individuals engage in various coping efforts or behaviors, that may either alleviate or exacerbate their sleep difficulties, such as increased caffeine consumption, or reliance on sleep aids (Folkman & Moskowitz, 2000; Folkman, 2013; Baltazar et al., 2019; Ben Simon et al., 2020). A coping effort refers to an individual’s attempt to regulate distress or manage a stressor, even if the effectiveness of that effort is uncertain or inconsistent (Folkman, 2013). These can include self-regulatory efforts, such as active efforts to regulate one’s own stress or arousal levels, for instance engaging in regular physical activity—as well as passive, avoidance-coping efforts intended to reduce or temporarily distract from distress (e.g., viewing television before sleep to distract from anxiety or stress) (Folkman & Moskowitz, 2000; Folkman, 2013; Baltazar et al., 2019). A cursory distinction between active and avoidance-coping efforts is that active coping efforts are assumed to directly address the underlying causes of distress, whereas the latter focuses on minimizing its effects (Folkman & Moskowitz, 2000).

These coping efforts often manifest in specific behavioral patterns. Whether these efforts are beneficial or maladaptive may depend on factors such as the individual’s stress levels, the nature of their sleep difficulties, or the effectiveness of the coping effort. On the one hand, an active coping effort might support higher-quality sleep by definitively reducing pre-sleep arousal (Folkman & Moskowitz, 2000; Folkman, 2013). On the other hand, avoidance-coping efforts may provide temporary relief from psychological distress but could contribute to persistent sleep difficulties if individuals fail to address the root causes of their arousal before sleep (Lund, 2021). For instance, Brown, Qin & Esmail (2017) found that students reporting the greatest sleep problems were also the most likely to use music before sleep as a coping-oriented effort.

Additional self-help factors: demographics and behavior

Alongside psychological distress, other demographic and behavioral factors may influence the association between self-help approaches and sleep. First, age and gender contribute to variations in sleep quality. Sleep architecture typically becomes lighter with age, and sleep disorders such as insomnia and sleep-disordered breathing are more prevalent in older adults (Ohayon, 2002; Miner & Kryger, 2017). Women may be disproportionately influenced by stress-related sleep disturbances, partly due to hormonal and psychosocial factors (Drake et al., 2004). As previously stated, younger adults are more likely to incorporate music as a sleep aid (Morin et al., 2006; Buus, Genovese & Jespersen, 2025), suggesting demographic variation in its adoption. Second, physical activity frequency (PAF) is consistently linked to improved sleep outcomes, with meta-analyses showing moderate improvements in sleep quality among physically active individuals (Baron, Reid & Zee, 2013; Kredlow et al., 2015). Recent work also highlights physical activity’s protective role in moderating the association between poor sleep quality and psychological distress, with (Zhang et al., 2022) reporting higher frequencies of physical activity buffering against the detrimental effects of poor sleep on daily functioning and distress among older adults. These benefits are often attributed to physiological regulation (e.g., circadian alignment, parasympathetic activation) and cumulative health effects, rather than immediate reductions in pre-sleep arousal (Kredlow et al., 2015). Thus, examining their combined associations provides an opportunity to explore whether distinct self-help behaviors interact to influence sleep quality. Cumulatively, PAF, age, and gender represent behavioral and demographic factors that may influence how self-help approaches associate with pre-sleep music use. While psychological distress provides the primary theoretical rationale for examining music as a sleep aid, these additional variables allow for an exploratory investigation of whether the association between music and sleep quality differs across behavioral or demographic subgroups.

Research gaps and study objectives

Few studies have examined how self-help behaviors, such as self-selected music use, interact in relation to psychological distress and sleep difficulties. The coping literature (Folkman & Moskowitz, 2000; Folkman, 2013) and the hyperarousal model (Bonnet & Arand, 2010; Riemann et al., 2010) indicate that music use may operate as a coping-oriented approach to downregulate pre-sleep arousal. However, it remains unclear whether individuals experiencing greater distress are more likely to use music, or whether music attenuates the distress-sleep association. In addition, although PAF, age, and gender are established correlates of sleep quality (Morin et al., 2006; Kredlow et al., 2015; Zhang et al., 2022; Buus, Genovese & Jespersen, 2025), little is known about their role alongside pre-sleep music use. Accordingly, the present study adopts an exploratory approach to address the following research questions:

RQ1. To what extent is the association between psychological distress (e.g., anxiety, mood disturbance, stress) and sleep quality related to the use of self-selected pre-sleep music?

Exploratory aim 1: To explore whether self-selected pre-sleep music is associated with variations in the relationship between psychological distress and reported sleep quality.

RQ2. Do demographic and behavioral factors (age, gender, and PAF) relate to how self-selected pre-sleep music use is associated with sleep quality?

Exploratory aim 2: To examine whether the associations between pre-sleep music use and sleep quality differ across subgroups defined by demographic or behavioral characteristics.

Materials and methods

Procedure

This study was designed as a cross-sectional online survey, to explore pre-sleep music use and sleep quality in adults. The cross-sectional design allowed for the explorative investigation of associations between pre-sleep music, physical activity, emotional states, and sleep quality in a sample of adults. The data sampling procedure was restricted to individuals who are fluent in the English language. The survey period ran from November 29, 2024, to January 31, 2025. Methodological and procedural information can be found in a preprint of an earlier version of this study (Danso et al., 2025).

Participants

A sample of 269 participants (Mage = 27.71, SD = 9, range: 18–82) was recruited from the internet and two social media platforms, Facebook and Reddit. The sample included 142 females (52.59%), 105 males (39.04%), and 22 gender-diverse individuals (8.37%). Participants had a mean age of 27.7 years (SD = 9.0, range = 18–79). The median age was 26 years. Participants reported their highest educational attainment as follows: 88 held a Bachelor’s degree, 51 had a Master’s degree, and 62 had completed high school. Regarding employment status, 102 participants were students, 119 were employed, and 19 were currently out of work. As these demographic items were optional, totals do not sum to the full sample size, reflecting missing responses. Participants from 45 countries responded to the survey. The majority identified as American (37.5%), Canadian (14.8%), or British (9.4%). A power analysis was informed by Muthén & Muthén (2002) simulation studies for detecting interaction effects in single-level structural models. A sample size of approximately 250 participants is generally adequate to detect small-to-moderate interaction effects (standardized β ≈ 0.15–0.20) using observed variables and robust estimation. With a sample size of N = 269, the study was sufficiently powered (≥80%) to detect interaction effects of this magnitude in a single-level path model.

Operationalization of self-selected pre-sleep music

Self-selected pre-sleep music was operationalized as music that participants independently chose to listen to prior to sleeping, characterizing personal preference and familiarity. Prior evidence indicates that preference and familiarity are central to music’s relaxation and anxiety-alleviating effects (Iwaki, Tanaka & Hori, 2003; Tan et al., 2012), and that such choices are highly idiosyncratic, spanning diverse genres and artists (Trahan et al., 2018). A full list of participant-reported music selections is provided in the Supplemental Files.

Materials

The online questionnaire was administered using the online survey tool REDCap (Research Electronic Data Capture, Harris et al., 2009) via social media platforms (Facebook and Reddit) and relevant mailing lists, directing participants to the anonymous survey, and required approximately 20 minutes to complete. The survey was explicitly advertised on the internet as a study on ‘Music and Sleep’. The questionnaire collected data on participants’ self-selected music for sleep, PAF, psychological distress (e.g., state anxiety, mood, and stress), and sleep quality. The study was conducted in accordance with the ethical principles of the National Advisory Board on Research Ethics in Finland (TENK; see https://www.tenk.fi/sites/tenk.fi/files/ethicalprinciples.pdf). Participants received a privacy notice in line with GDPR, provided informed consent prior to participation, and were informed of their rights (e.g., withdrawal, data protection). Informed consent was obtained electronically: participants read the privacy notice and research notification and then confirmed their consent by clicking an “I agree to participate in this study” button on the computer screen, before accessing the questionnaire. Furthermore, no direct identifiers were collected, and responses were anonymized at the compilation stage.

The questionnaire included measures of the following constructs: Demographic information: Participants provided details on their age, gender, nationality, highest educational degree, and current occupation.

Pittsburgh Sleep Quality Index (PSQI): The PSQI is a 19-item self-report questionnaire designed to assess sleep quality disturbances over a 1-month interval. These items generate seven component scores: subjective sleep quality, sleep latency, sleep duration, habitual sleep efficiency, sleep disturbances, use of sleeping mediation, and daytime dysfunction. The sum of these component scores provides a global score ranging from 0 to 21, with higher scores indicating poorer sleep quality. The original scale is widely recognized for strong internal consistency α = 0.83, as well as diagnostic sensitivity (89.6%) and specificity (86.5%) in distinguishing between good and poor sleepers (Buysse et al., 1989).

Self-selected pre-sleep music item: To assess the use of music before sleep we incorporated a custom item into the standardized PSQI, asking participants: “During the past month, have you used music to help you sleep?” Open-ended responses were coded onto a 5-point Likert scale ranging from 1 (Never) to 5 (Always). Although this item does not explicitly specify whether the music was self-selected, we assumed that participants were reporting their own music choices. Nevertheless, the lack of specificity limits interpretability regarding the self-selected aspect of music use. To validate this measure, two independent raters coded the responses, resulting in strong inter-rater reliability of k = 0.92, supporting its use as an indicator of pre-sleep music use in our analysis. This approach was used to offer an unambiguous operational definition of music use for sleep, which facilitates comparability with prior research (Trahan et al., 2018) using the PSQI.

To be able to comprehensively understand the relationship between pre-sleep music and sleep quality, we additionally included four open-ended descriptive responses in the survey. These included: “Why do you believe music helps you sleep?”, “Why haven’t you used music to help you sleep?”, “If you listen to music before sleep, how does it affect your ability to fall asleep?”, “How does listening to music affect your mood and anxiety levels?”

Sports and Physical Activity Questionnaire: The Sports and Physical Activity Questionnaire is based on the Special Eurobarometer survey conducted by the European Commission and was used as an indicator of participants’ PAF. It includes a single primary question that assesses the frequency of sports and physical activity: “How often do you exercise or play sport?”. Respondents chose from six categories: (1) “5 times a week or more,” (2) “3 to 4 times a week,” (3) “1 to 2 times a week,” (4) “1 to 3 times a month,” and (5) “less often”. For analysis, PAF was treated as a continuous numeric variable, ensuring that variance in activity frequency was preserved rather than dichotomized (e.g., regularly active vs. inactive). This questionnaire has been validated through its use in large-scale, representative surveys across European Union countries, demonstrating reliability in capturing cross-national variations in physical activity frequencies (European Commission, Directorate General for Education and Culture, & TNS Opinion & Social, 2014).

State anxiety: Anxiety was measured using the state anxiety subscale (STAI-S) of the State-Trait Anxiety Inventory (STAI, Spielberger et al., 2017). The STAI-S consists of 20 items assessing current situational anxiety (e.g., “I feel tense”), each rated on a 4-point Likert scale. Total scores range from 20 to 80, with higher scores indicating greater state anxiety. The STAI-S has demonstrated excellent internal consistency in prior research (α = 0.86–0.95). In our data, α was 0.94.

Profile of Mood States-Short Form (POMS-SF): The POMS-SF is a condensed version of the original profile of mood states short (POMS) questionnaire and includes 37 items that measure six mood dimensions: tension-anxiety, depression-dejection, anger-hostility, vigor-activity, fatigue-inertia, and confusion-bewilderment. Each item is rated on a 5-point Likert scale from “Not at all” to “Extremely”. Validation studies report high internal consistency across subscales α > 0.90 and strong correlations with the full-length version (Shacham, 1983). In our data, α was 0.94.

Perceived Stress Scale (PSS): The PSS is a widely used psychological instrument consisting of 10 items that assess perceived stress levels over the past month. Items are rated on a 5-point Likert scale from “Never” to “Very often,” with total scores ranging from 0 to 40; higher scores indicate greater perceived stress. The scale has demonstrated strong internal consistency, α = 0.78–0.91 and construct validity through significant correlations with measures of anxiety and depression (Cohen, Kamarck & Mermelstein, 1983). In our data, α was 0.91.

Statistical analyses

To address our research questions, we specified two path models using the lavaan package in R (Rosseel, 2012) to examine (1) how self-selected pre-sleep music use was associated with the relationships between psychological distress variables (state anxiety, mood disturbance, and stress) and sleep quality (RQ1), and (2) whether demographic and behavioral factors (PAF, age, and gender) were associated with variations in the music-sleep relationship (RQ2). Interaction terms between music use and each predictor were specified to explore these associations. State anxiety, mood disturbance, and stress were included as covariates in the model, with correlations specified among these predictors to account for their interrelations. Model fit was assessed using standard indices, including the Comparative Fit Index (CFI), Tucker-Lewis Index (TLI), root mean square error of approximation (RMSEA), and standardized root mean square residual (SRMR), with global sleep quality (PSQI score) specified as the outcome. The final model accounted for approximately 40% of the variance in sleep quality (R2 = 0.40), indicating a moderate to strong explanatory effect.

Results

Descriptive statistics and correlations

Participants reported an average PSQI score of 6.75 (SD = 3.99), indicating generally poor sleep quality in the sample. Pre-sleep music use had a moderate score (M = 2.11, SD = 1.29, range: 1–5), suggesting variability in habitual engagement with music before sleep. Sleep quality was not significantly correlated with age (r = –0.09, p = 0.14), and mean PSQI scores did not differ significantly across gender groups, F(3, 251) = 0.53, p = 0.66 (male: M = 7.17, SD = 3.95; female: M = 6.67, SD = 4.10; diverse: M = 6.81, SD = 3.76; other: M = 5.25, SD = 4.50). Table 1 presents the descriptive statistics for key study variables, including sleep quality (PSQI), pre-sleep music use, state anxiety (STAI-S), mood (POMS-SF), stress (PSS), and PAF.

Table 1 Descriptive statistics for PSQI, self-selected pre-sleep music, STAI-S, POMS-SF, PSS and physical activity frequency.

Survey items	M	SD	Min	Max	
PSQIA	6.75	3.99	0	19	
Self-selected pre-sleep musicB	2.11	1.29	1	5	
STAI-SC	43.8	13.14	21	75	
POMS-SFD	29.61	33.15	−7	132	
PSSE	12.2	11.67	0	38	
PAFF	3.38	1.34	1	5	
Notes:

A–FPresents the theoretical minimum and maximum scale ranges:

A PSQI: Pittsburgh Sleep Quality Index (0–21).

B Self-selected pre-sleep music (1–5).

C STAI-S: State anxiety subscale (20–80).

D POMS-SF: Profile of Mood States–Short Form (−28 to 140).

E PSS: Perceived Stress Scale (0–40).

F PAF: Physical activity frequency (1–5).

Pre-sleep music use was positively associated with poorer sleep quality (r = 0.23, p < 0.01). Sleep quality was also correlated with higher state anxiety (r = 0.47, p < 0.01), mood disturbance (r = 0.55, p < 0.01), and stress (r = 0.56, p < 0.01). Physical activity was negatively correlated with PSQI scores (r = −0.15, p = 0.041). However, physical activity was not significantly associated with pre-sleep music use (r = −0.03, ns). Table 2 displays bivariate correlations among study variables.

Table 2 Correlations between PSQI, self-selected pre-sleep music, STAI-S, POMS-SF, PSS and physical activity frequency (N = 269).

Asterisks indicate that the values are Pearson correlations (two-tailed).

Survey item	PSQI	Self-selected pre-sleep music	STAI	POMS-SF	PSS	PAF	
PSQIA	1						
Self-selected pre-sleep music	0.23**	1					
STAI-SB	0.47**	0.15+	1				
POMS-SFC	0.55**	0.13+	0.7**	1			
PSSD	0.56**	0.16*	0.55**	0.87**	1		
PAFE	−0.15*	−0.03	−0.26	−0.17	−0.16*	1	
Notes:

A PSQI: Pittsburgh Sleep Quality Index.

B STAI-S: State anxiety subscale.

C POMS-SF: Profile of Mood States-Short Form.

D PSS: Perceived Stress Scale.

E PAF: Physical activity frequency.

+ p < 0.10.

* p < 0.05.

** p < 0.01.

RQ1. Exploring how pre-sleep music use relates to the association between distress and sleep quality

Higher frequency of pre-sleep music use was marginally associated with poorer sleep quality (β = 0.103, p = 0.086). State anxiety (β = 0.295, p < 0.001) and stress (β = 0.313, p = 0.001) also appeared significantly associated with poorer sleep quality. Mood disturbance (β = −0.038, p = 0.740) and physical activity (β = −0.027, p = 0.510) were not significant predictors in the model. The model accounted for approximately 40% of the variance in sleep quality (R2 = 0.396).

Direct associations

The interaction term between state anxiety and music listening was borderline (β = −0.170, p = 0.050). Simple slope analyses (Fig. 1) indicated that state anxiety was associated with poorer sleep quality across all levels of music listening frequency. Specifically, the association was strongest at low music listening frequency (−1 SD; β = 0.408, SE = 0.111, 95% CI [0.190–0.627]), somewhat weaker at medium frequency (β = 0.337, SE = 0.092, 95% CI [0.156–0.518]), and weakest at high frequency (β = 0.266, SE = 0.114, 95% CI [0.041–0.491]). No other moderation effects were observed, including interactions with mood (β = 0.134, p = 0.209), stress (β = −0.095, p = 0.280), or physical activity (β = 0.055, p = 0.182).

Figure 1 Interaction between state anxiety and music listening frequency on poor sleep.

Association between state anxiety and sleep quality at different levels of music listening frequency. Across all levels, higher state anxiety was linked with poorer sleep quality, with the association appearing strongest at low music listening frequency.

RQ2. Exploring whether demographic and behavioral factors relate to the music-sleep association

To examine whether demographic and behavioral factors (age, gender, physical activity frequency) moderate the effects of self-selected pre-sleep music on sleep quality, we conducted an additional moderation analysis. Due to the model being saturated, model fit indices (e.g., χ2, CFI, RMSEA) were deemed not informative and are therefore not reported.

Subgroup patterns

The results are displayed in Table 3. They indicated that the direct effect of self-selected pre-sleep music on sleep quality was borderline (β = 0.193, p = 0.073) when controlling for age, gender and physical activity frequency. However, age, gender, and physical activity were not significantly associated with global sleep quality. Moreover, the interaction terms with pre-sleep music for age, gender, and physical activity were not statistically significant, suggesting that these demographic and behavioral factors do not moderate the effect on sleep quality.

Table 3 Moderation analysis: direct and interaction effects of pre-sleep music, age, gender, and physical activity on sleep quality.

Predictor	ß	SE	p	
Pre-sleep music → Sleep quality	0.193	0.11	0.073	
Age → Sleep quality	0.033	0.64	0.610	
Gender → Sleep quality	0.126	0.12	0.303	
Physical activity → Sleep quality	−0.105	0.08	0.168	
Pre-sleep music × Age	−0.069	0.08	0.383	
Pre-sleep music × Gender	0.018	0.14	0.899	
Pre-sleep music × Physical activity	−0.061	0.07	0.398	

Discussion

Preliminary findings

The present exploratory study investigated patterns of self-selected pre-sleep music use in relation to sleep quality and psychological distress. Overall, pre-sleep music use was modestly associated with poorer reported sleep quality. A borderline interaction (p = 0.050) between state anxiety and music use presented a pattern that the anxiety-sleep association may be somewhat weaker among individuals who reported more frequent music use. This tentative pattern should be interpreted cautiously, as the cross-sectional design and marginal significance prevent any substantial conclusions. No moderation effects were observed for mood, stress, age, gender, or physical activity, indicating that any potential role of pre-sleep music was not strongly differentiated across these subgroups.

Our findings add to prior work showing that many individuals incorporate music into their pre-sleep routines (Morin et al., 2006; Trahan et al., 2018; Lund, 2021; Jespersen et al., 2022; Buus, Genovese & Jespersen, 2025). Aligned with survey evidence that frequent music users often report greater sleep difficulties (Brown, Qin & Esmail, 2017; Buus, Genovese & Jespersen, 2025), the present data showed that pre-sleep music use was modestly associated with poorer sleep quality. This pattern may reflect music being adopted reactively as a coping effort in response to distress (Folkman, 2013; Folkman & Moskowitz, 2000). In line with prior evidence that music can down-regulate arousal and reduce state anxiety (Iwaki, Tanaka & Hori, 2003; Tan et al., 2012; Trahan et al., 2018), we observed a borderline interaction that could potentially present as a tentative buffering effect of music used among more anxious individuals.

Thus, given that anxiety is a well-established predictor of sleep difficulties (Harvey, 2002; Bonnet & Arand, 2010; Riemann et al., 2010), this exploratory result may be somewhat descriptive of music listening used as a coping-oriented effort that individuals adopt to regulate pre-sleep psychological distress. This observed pattern, in which individuals experiencing greater psychological distress reported both poorer sleep and more frequent music use, may tentatively reflect a coping approach to minimize the influence of state anxiety before sleep, consistent with prior conceptualizations of music used as a coping effort to manage pre-sleep difficulties (e.g., Folkman & Moskowitz, 2000; Folkman, 2013). With respect to Folkman’s coping framework (Folkman & Moskowitz, 2000; Folkman, 2013), we speculate that our sample used music as a context-sensitive (i.e., situational) self-help approach to minimize state anxiety prior to sleep, influenced by the individual’s appraisal of music’s controllability (e.g., choosing music when other stressors feel uncontrollable) and coping tendencies (e.g., a tendency to use emotion-focused strategies such as distraction). Importantly, this interpretation remains speculative given the cross-sectional and correlational nature of the data but is also in line with the assumption that healthy sleepers require little assistance to initiate sleep (Bjorvatn, Waage & Saxvig, 2023).

Although state anxiety emerged as the most relevant distress variable in the analyses, the inclusion of stress and mood disturbance provided a nuanced picture. Stress predicted poorer sleep quality but was not moderated by music use, whereas mood disturbance did not emerge as a significant predictor. This partially aligns with prior evidence that distress is a correlate of sleep difficulties (Brown, Qin & Esmail, 2017; Blackwelder, Hoskins & Huber, 2021) but suggests that music’s potential buffering role may be more specific to state anxiety-related arousal. Within the hyperarousal model (Bonnet & Arand, 2010; Riemann et al., 2010), psychological factors such as state anxiety are implicated in the persistence of sleep disturbances, and the present results may be viewed in light of this framework. Further, the positive association between music use and poorer sleep may describe individuals with heightened distress adopting behavioral patterns to manage pre-sleep arousal. Importantly, while hyperarousal frameworks largely characterize such behaviors as maladaptive perpetuating mechanisms, the observed borderline moderation effect leaves open the possibility that music use could, in some cases, function as a self-help coping effort that attenuates the association between state anxiety and poor sleep. In line with the coping effort conceptualization (Folkman & Moskowitz, 2000; Folkman, 2013), such approaches may provide temporary regulation of distress but do not address the underlying causes of sleep difficulties.

In our exploratory analyses of demographic and behavioral factors, age, gender, and physical activity did not significantly moderate the association between pre-sleep music use and sleep quality. The relatively high prevalence of pre-sleep music use in our young adult sample aligns with prior survey research indicating that younger individuals are more likely to use music as a sleep aid (Morin et al., 2006; Buus, Genovese & Jespersen, 2025). However, despite prior survey research indicating that younger adults are more likely to use music as a sleep aid (Morin et al., 2006; Buus, Genovese & Jespersen, 2025), our results may suggest that the observed associations between music use, state anxiety, and sleep quality were not age dependent. Overall, the associations between music use, distress, and sleep quality appeared relatively consistent across demographic and behavioral subgroups. While regular physical activity is frequently associated with improved sleep outcomes (Baron, Reid & Zee, 2013; Kredlow et al., 2015), in the present study, PAF somewhat correlated with better sleep quality, but was not a significant predictor in regression models, and our findings did not indicate any additive benefits when music use and physical activity were considered together. Instead, physical activity may exert its influence on sleep quality largely independently of music use (and vice-versa), suggesting distinct underlying mechanisms.

As noted by Kredlow et al. (2015), physical activity frequency is generally associated with cumulative physiological benefits that accrue over time relative to sleep, whereas music use appears to influence sleep more immediately through avoidance-coping efforts in response to pre-sleep distress (Folkman & Moskowitz, 2000; Folkman, 2013). Although age and gender are well-documented correlates of sleep quality (Drake et al., 2004; Morin et al., 2006; Miner & Kryger, 2017; Buus, Genovese & Jespersen, 2025), their lack of interaction with music use in this sample suggests that music’s role as a self-help approach is not strongly differentiated across demographic groups. Furthermore, while prior research has suggested that women may be more sensitive to stress-induced sleep disruptions (Drake et al., 2004), and that age can influence sleep quality (Ohayon, 2002; Zhang et al., 2022), the present data did not indicate these variables to moderate the relationship between pre-sleep music use and sleep quality.

Limitations

A key limitation of this study is its exploratory, cross-sectional design, which precludes causal inference, limiting the strength of conclusions that can be drawn from marginal effects. Thus, we cannot determine the directionality of the association between pre-sleep music use and poor sleep quality, for instance, whether individuals with greater distress turn to music in response to sleep problems, or whether music use itself reflects a broader pattern of coping efforts. Experimental or longitudinal studies are needed to establish temporal order (e.g., the temporal precedence of variables) and clarify the conditions under which music use may be most effective. For instance, longitudinal or experimental designs may examine whether pre-sleep music use prospectively moderates the influence of psychological distress, helping to establish temporal precedence. Furthermore, models may explore conditional effects across subgroups (e.g., high vs. low state anxiety, physically active vs. inactive individuals), to determine for whom and under what conditions music use is most beneficial. Given the modest sample size and the borderline significance of the interaction, these results should be interpreted as preliminary and hypothesis-generating; replication in larger and more adequately powered samples will be essential to determine whether the observed moderation effect is robust (Wolf et al., 2013).

Additionally, our study relied on self-reported sleep quality (PSQI) and self-reported music use frequency, which are subject to recall bias and perceptual distortions. The PSQI captures perceived sleep quality rather than objective physiological sleep parameters (e.g., sleep efficiency, rapid eye movement (REM) duration). While our PSQI custom item provides a standardized way to identify music users, it also does not account for individual differences in how music is selected or experienced, which may influence its efficacy as a sleep aid. Music used for sleep may differ in how it is used, whether to promote relaxation, mask noise, or simply as a habitual sleeping behavior (Lee-Harris et al., 2018; Dickson & Schubert, 2022; Scarratt et al., 2023; Kirk & Timmers, 2024). Some individuals may listen before sleep, while others may listen throughout the night, which may result in different effects on sleep outcomes. Future research should therefore investigate how listening contexts and functions (e.g., relaxation, noise masking, habit) influence sleep quality rather than treating music use as a uniform behavior.

Pre-sleep music use was assessed with an open-ended question that was subsequently coded into a 5-point scale ranging from ‘Never’ to ‘Always’. Although this provided a measure of frequency, the coding may have introduced variability and the response labels lack the specificity of time-based frequency formats (e.g., ‘once or twice a month’, ‘5–7 nights per week’). Such formats, used in previous studies (e.g., Trahan et al., 2018), would improve interpretability and comparability. Future research should therefore adopt validated, time-specific frequency measures to improve reliability. Furthermore, the measure of pre-sleep music use did not explicitly capture whether the music was self-selected. While it is plausible that participants typically reported their own music choices, this cannot be confirmed. Future studies should utilize validated, frequency-based measures that explicitly differentiate between self-selected and externally provided music. Notably, the survey was introduced as a study on ‘Music and Sleep,’ and the framing may have influenced participation, potentially attracting individuals with greater sleep difficulties, or higher music engagement, thereby limiting the representativeness of the broader adult population.

Finally, uncontrolled confounds such as sleep routines, environmental factors (e.g. ambient noise, temperature), and personality traits may have also influenced both music use and sleep quality. Future studies could address these limitations by incorporating standardized sleep diaries to track bedtime routines, objective sleep tracking, and inclusion of relevant covariates, thereby providing a more precise picture of how pre-sleep music relates to sleep outcomes.

Conclusions

The present study found that individuals reporting poorer sleep quality were more likely to engage in pre-sleep music listening. Moderation analyses indicated a borderline pattern in which the association between state anxiety and poor sleep quality appeared strongest among those who seldom used music and somewhat weaker among more frequent users. This tentative pattern did not vary by age, gender, or physical activity frequency. Given the cross-sectional design, causal inferences cannot be drawn; accordingly, these findings should be viewed as preliminary and hypothesis-generating. Nonetheless, the results may inform understanding of pre-sleep music use as a potential self-help approach that some individuals employ to manage state anxiety prior to sleep, even though it was not associated with better overall sleep quality. Specifically, the observed associations align with prior research on psychological distress and coping efforts in sleep regulation, as state anxiety was associated with a portion of sleep difficulties, highlighting the relevance of cognitive-emotional arousal in poor sleep quality. This interpretation aligns with self-regulatory and avoidance-coping frameworks, which imply that individuals may engage in efforts, such as music listening, to mitigate psychological distress prior to sleep. With respect to practical implications, the findings suggest that while pre-sleep music listening may provide situational relief from state anxiety-related arousal, directly addressing underlying psychological distress is likely to yield sustained improvements in sleep quality than reliance on music alone. Future research may prioritize longitudinal or experimental designs to better clarify causal relationships, and explore whether music characteristics (e.g., tempo, lyrics) or listening contexts (e.g., duration, timing) are associated with differences in sleep outcomes.

Supplemental Information

Supplemental Information 1 Participant-reported music selections.

Records of each participants responses to survey item “What music do you listen to sleep?”.

Supplemental Information 2 Syntax for research questions 1 & 2.

Supplemental Information 3 Raw dataset.

The authors wish to acknowledge Dr. Alessandro Ansani for his valuable assistance with data analysis.

Additional Information and Declarations

Competing Interests

The authors declare that they have no competing interests.

Author Contributions

Andrew Danso conceived and designed the experiments, performed the experiments, analyzed the data, authored or reviewed drafts of the article, and approved the final draft.

Mareike Ehlert conceived and designed the experiments, performed the experiments, analyzed the data, prepared figures and/or tables, authored or reviewed drafts of the article, and approved the final draft.

Friederike Koehler conceived and designed the experiments, authored or reviewed drafts of the article, and approved the final draft.

Rory Kirk conceived and designed the experiments, authored or reviewed drafts of the article, and approved the final draft.

Nandhini Natarajan conceived and designed the experiments, authored or reviewed drafts of the article, and approved the final draft.

Shannon Eilyce Wright conceived and designed the experiments, authored or reviewed drafts of the article, and approved the final draft.

Renee Timmers conceived and designed the experiments, authored or reviewed drafts of the article, and approved the final draft.

Suvi Saarikallio conceived and designed the experiments, authored or reviewed drafts of the article, and approved the final draft.

Human Ethics

The following information was supplied relating to ethical approvals (i.e. approving body and any reference numbers):

University of Jyväskylä, Department of Music, Art, and Culture Studies.

Data Availability

The following information was supplied regarding data availability:

The data and syntax are available in the Supplemental Files.

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
