# Peer review of "Patterns of pre-sleep music use and sleep quality: exploratory survey findings on state anxiety"

_PeerJ, doi:10.7717/peerj.20444_

## Round 0.1 · original submission · Major Revisions

· Academic Editor

Major Revisions

·

Basic reporting

Danso et al present results from a small online survey to investigate whether the link between anxiety and subjectively rated sleep quality is moderated by self-reported music listening before sleep, the assumption being that at high levels of anxiety, music may have a beneficial effect on sleep quality. While this research question is interesting and potentially relevant, the design of the study and the interpretation of the results are, in my view, inadequate to answer it.

Experimental design

The study relies on a cross-sectional, online sample with anxiety, stress, music listening, and sleep subjectively rated in a single session. This design is not well-suited to answer the research question, because it is difficult to disentangle trait-level variables and response styles from the target phenotype of the questionnaire. Ideally, the authors would use a multi-day study where they would investigate if the same person reports better sleep after listening to music than otherwise, and if this relationship is moderated by trait anxiety. Objective measures of sleep would enhance this design further, as subjectively rated sleep is known to be weakly correlated to objective measures and arguably better with negative emotionality.

Validity of the findings

The authors find a “marginally significant” effect in the analysis testing their hypothesis: an interaction p-value of exactly 0.05. I disagree with the strategy of the authors to interpret this as tentative support for their hypothesis. Borderline p-values, especially for interaction effects, are associated with poor replicability. In my view, the best interpretation is a null finding; however, it is difficult to say if this is because of an actually absent relationship or just low power. Given that the study is based on an online survey in English about common phenotypes, a sample size of just 269 is hard to justify.

Additional comments

Because of these reasons, I cannot support the publication of this particular manuscript.

·

Basic reporting

The study investigates whether pre-sleep music use moderates the relationship between anxiety and poor sleep using online survey data from 269 participants. Furthermore, the role of individual differences such as physical activity, age, and gender is investigated as variables moderating the impact of music on sleep quality. The research question is interesting, but the reporting suffers from a lack of clarity in several ways. Overall, he introduction is not clearly structured and does not build a clear rationale for the hypotheses, and the literature on music and sleep referenced is limited and does not include recent studies to a sufficient degree.

The introduction starts with a focus on insomnia and insomnia treatment, but lacks information on self-help strategies for improving sleep, which seems to be more relevant to the study and is reported in several studies (e.g., Morin et al., 2006). Epidemiology of insomnia: Prevalence, self-help treatments, consultations, and determinants of help-seeking behaviors. Sleep Medicine, 7(2), 123-130; Brown et al. (2017). “Sleep? Maybe Later…” A Cross-Campus Survey of University Students and Sleep Practices. Education Sciences, 7(3), 66. Bjorvatn et al. (2023). Do people use methods or tricks to fall asleep? A comparison between people with and without chronic insomnia. Journal of Sleep Research, 32(2), e13763.; Buus et al. (2025). The art of sleep: examining sleep strategies in the general population with a focus on the use of music for sleep. Journal of Sleep Research, e70006.

Please be careful not to conflate sleep difficulties and insomnia disorder. Insomnia symptoms are not assessed in the current study, and hence, I recommend focusing on sleep difficulties in general and not in a diagnostic sense, such as insomnia disorder. Also, if insomnia treatment is included in the introduction, it needs to be more precise. The limitations of medical treatment and cognitive behavioral therapy for insomnia are very different and cannot be grouped.

An important point is that a substantial amount of relevant literature on the topic of music and sleep is not included. The distinction between the use of music for sleep and the effect of music on sleep is not clear from the beginning, nor in the way many statements are referenced. Studies evaluating the effects of music on sleep exist, but are missing from the manuscript or are not used actively. For example, the effect of music as a sleep aid is referenced with two relatively old studies (ll. 116-118). Several more recent meta-analyses based on a larger number of studies exist and should be included. In addition, since the study focuses on how pre-sleep music use may moderate the association between anxiety and sleep quality, the studies reporting on the suggested mechanisms underlying the impact of music on sleep should be given more emphasis. Mainly, the Trahan 2018 study is used as a reference, but other studies also present relevant information (e.g., Dickson, G. T., & Schubert, E., 2019). How Does Music Aid Sleep? Literature Review. Sleep Medicine.; Jespersen, K. V. (2022). A Lullaby to the Brain: The Use of Music as a Sleep Aid. In B. Colombo (Ed.), The Musical Neurons (pp. 53-63); Lund et al. (2021). Music, sleep, and depression: An interview study. Psychology of Music, 03057356211024350).

Similarly, recent studies on the use of music as a self-selected strategy for sleep improvement do exist, but are not included. As such, the study does not seem sufficiently rooted in existing knowledge of the topic. In some cases, the use of references is misleading, e.g., Introduction, ll. 112-114: “While music-assisted relaxation for sleep has received recognition for its potential to improve sleep through self-reported relaxation and physiological regulation (e.g., lowering heart rate and lowering arousal)”. Only one of the studies references actually measures arousal (self-reported), and none of the studies include heart rate measurements. Hence, in the current state, the references used do not always support the claims of the text. Similarly, the following sentence (ll. 120-122) is referenced for content about music and sleep, but the reference is only on music for relaxation, and sleep is not mentioned in the paper. Please make sure to use references specific to the given arguments.

This limited use of recent music and sleep literature is also present in the discussion, e.g., ll. 367-370. I don’t find that recent research presents a simplified universal framing of music as a non-pharmacological aid for sleep improvement. In contrast, I think that a wide variety of mechanisms at various levels have been discussed in the literature (see references above). Please do a thorough literature review on music and sleep studies, and include relevant references to strengthen both the introduction and discussion.

Please report information on the distribution of the survey. The way the survey is framed (e.g., sleep survey, music survey, music and sleep survey, physical activity survey) may influence the sample characteristics, e.g., sleep surveys often attract people with poor sleep, and music surveys often attract people with high levels of music engagement. As such, this information is important. Please also report descriptives on how sleep quality is related to age and gender to clarify the characteristics of the sample and how these align with previous knowledge on sleep quality, age, and gender.

The title is long and not very clear. Please provide a title that conveys the message of the study in a way that is more accessible to the reader.

It is stated that the study was exempt from ethical approval; however, the supplemental documents show that the study documents were reviewed and deemed without ethical problems. Furthermore, the documents also show that participants gave signed consent. Please specify these aspects in the text.

Experimental design

The research question is relevant, but the argumentation for the hypotheses is not well presented. The introduction is not clearly structured and does not build a solid argument for the relevance of the study. Since the focus of the study is on the relationship between psychological distress and sleep quality, this topic should be clearly reviewed. For example, the moderating effect of pre-sleep music use on the relationship between anxiety and poor sleep is investigated, but one could also argue to investigate the moderating effect of anxiety on the effect of music on sleep quality. Hence, the argument for the choice of focus needs to be clear. Furthermore, the argument why physical activity should moderate the effect of pre-sleep music on sleep is not clear from the introduction. Both music and exercise can serve as self-help strategies to improve sleep, but the reason for investigating the moderating role of exercise on pre-sleep music use needs to be rooted in clear argumentation. In addition, there is no clear argumentation for the role of age and gender as moderating variables on the relationship between the use of pre-sleep music and sleep quality. Hence, Hypothesis 2 does not seem substantiated. The introduction should provide relevant information and build a clear argument for the hypotheses of the study.

The study finding is a marginally significant effect of pre-sleep music use moderating the relationship between anxiety and sleep quality. No other parts of the hypotheses were supported. If the hypotheses were strong and well-argued, all findings would be interesting, but since the rationale for the hypotheses is not clearly argued in the introduction, the results become less meaningful to the reader.

Methodologically, it doesn’t seem optimal that participants gave open-ended responses that were then coded to a 5-point Likert scale by the researchers. Why did participants not respond on a Likert scale? The coding could introduce unnecessary noise in an important variable of the study. The item question was “During the past month, have you used music to help you sleep?” What if people just answer ‘Yes’? How should that be coded? This is not clear from the description. When asking about pre-sleep music use in the previous month, the data quality would have been better with more specific response options (e.g., average number of days per week). Please consider this in the limitations section.

It is not entirely clear if the study is hypothesis-driven (ll. 169-181) or explorative (ll. 184-187). Please clarify.

In the results section, it is not clear if Anxiety refers to trait or state anxiety. Please clarify. This also makes a difference for the interpretation of the results.

Validity of the findings

As stated above, the study has some limitations in its current version.
Data on the study have been provided in sufficient detail.

Reviewer 3 ·

Basic reporting

The manuscript is overall clean and readable. However, there are a few minor typographical mistakes that should be addressed:
- Line 72: Period instead of comma between ‘sleep’ and ‘including’.
- Line 90: Referred to avoidance-coping efforts, it says ‘focuses’ instead of ‘focus’.
- Line 193: ‘Diverse-gender’ is used rather than ‘gender-diverse’.
- Line 244: Missing capitalisation of ‘Physical Activity Level’.
- Line 410: Reference is put into brackets despite being included in the text.

Moreover, some sentences could benefit from some rephrasing for clarity and fluency, for example:
- Lines 22-23: ‘(a) whether self-selected pre-sleep music moderates the relationship between psychological distress (anxiety, mood, and stress)’ --> it is unclear here what psychological distress is being related to. Please clarify what the relationship is between.
- Lines 53-55: ‘Age-related changes in sleep architecture further contribute to sleep disturbances in older adults, including increased time in lighter sleep (Stage 1) and reduced time in deep sleep (Stages 3 and 4) (Ohayon, 2002; Miner & Kryger, 2017).’ --> This appears to refer to an outdated classification system for sleep stages. I recommend using the current AASM classification system (i.e., N1, N2, N3, REM), in which stages 3 and 4 are combined into N3.
- Lines 61-62 and line 85: ‘(e.g., medication, cognitive behavioral therapy) --> given the manuscript’s specific focus on sleep, these examples could be more specific, e.g., mentioning benzodiazepines and Cognitive-Behavioural Therapy for Insomnia (CBT-I).
- Lines 97-100: ‘On the other hand, avoidance-coping efforts may provide temporary relief from distress, and could contribute to persistent sleep difficulties, if individuals fail to address the root causes of their arousal before sleep (Lund, Helle Nystrup, 2021) --> the use of ‘but’ rather than ‘and’ before ‘can contribute to persistent sleep difficulties’ may enhance the flow of the sentence.

The article includes relevant literature in the introduction section. However, most cited studies date from 2000 to 2015. A more balanced inclusion of recent work could strengthen the theoretical background and context.

The in-text references should be checked for consistency, as they at times do not appear to follow the same format throughout the manuscript.
Overall, the manuscript follows a standard structure and it’s self-contained.

Experimental design

There are some aspects of the experimental design that merit further consideration.

First, regarding the hypotheses as presented in the ‘Research Gaps and Study Objectives’ section in the Introduction:

Given the importance in the literature on structured music interventions for sleep, it might be relevant to begin by exploring the direct effects of self-selected music on sleep. This would lay the ground to later delve into the role of self-selected music in the relationship between psychological distress and sleep quality.

In the ‘Participants’ section (‘Materials and Methods’, lines 194 – 196), the distribution of participants across categories of education and employment does not appear to cover the entire sample. It would be helpful to clarify this point (e.g., missing data due to non-compulsory questions).

The addition of a custom item to the PSQI provides a useful indication of how frequently participants used music to help them sleep. However, the item does not appear to capture the nature of the music used, nor whether it was personally selected by the participant – this music could be selected using an app, or else the participant could use random pre-existing playlists containing music that has not been selected personally. Since the concept of self-selected music seems to be central to the study’s theoretical framing (H1), a clearer operational definition of what is considered self-selected in this context would help to avoid ambiguity. This could help understand whether the added measure reflects that construct.

The Likert scale used ranges from ‘Never’ to ‘Always’, which provides a measure of frequency. However, using more time-specific labels (e.g., ‘Less than once a year’, ‘once or twice a month’, …) could aid interpretability and comparability. This approach would also align with previous research, such as Trahan et al. (2018), which is cited in the description of the added item. While this adjustment cannot be applied to the current data, it could be considered in future research designs.

Validity of the findings

As mentioned in the previous section, while the study addressed H1, concerning the moderating effects of self-selected music, it is unclear whether the PSQI item used to address the use of pre-sleep music adequately captures the self-selected aspect of the music. Thus, this limits the interpretability of the findings related to H1.

When coming to the implications of the study, the suggestion that directly assessing psychological distress is more beneficial for improving sleep quality than solely relying on pre-sleep music may come across as self-evident or overly general. It could be beneficial to clarify this point or to provide more nuanced implications.

---

## Round 0.2 · Minor Revisions

· Academic Editor

Minor Revisions

·

Basic reporting

Thanks to the authors for their thorough revision of the manuscript. The manuscript has improved substantially to present the line of argumentation and the results in a clear and coherent way.

Experimental design

Thanks for clarifying the exploratory nature of the design

Validity of the findings

-

Reviewer 3 ·

Basic reporting

This cross-sectional study aims to explore whether the use of self-selected pre-sleep music has an impact on the relationship between sleep quality and psychological distress. Moreover, it explores whether specific demographic or behavioral characteristics influence the association between self-selected pre-sleep music and sleep quality. For this, data from 269 participants collected through an online survey were used. In the survey, demographic information, psychological distress data (state anxiety, mood, and stress), sleep quality data, physical activity frequency, information about pre-sleep music, and specific information about the use of pre-sleep music have been collected.

The reframing of the study with an exploratory approach to address the abovementioned research questions gives a clearer and useful preliminary and hypothesis-generating view on the topic.
The introduction is now based on recent findings and gives a good context for the framing of the study. This new structure of the introduction gives a comprehensive and structured context.

Adding on the materials and on the ethical side of the data collection makes the methodology of the study more understandable.

Also, the ‘discussion’ and ‘limitations’ sections are now rephrased in a way that is in line with the introduction and with the aim of the study, and the result is more coherent.

The title is now more understandable and conveys the aim of the study more clearly.
A few minor comments on the English used in the paper:
- L143 – 146 The use of commas sounds slightly off. Maybe better ‘A cursory distinction between active and avoidance-coping efforts is that active coping efforts are assumed to directly address the underlying causes of distress, whereas the latter focuses on minimizing its effects’
- L235 ‘operationalized as music that participants independently chose to listen to prior to sleep-in’, there is a not needed ‘at’

Experimental design

As mentioned above, the reframing of the study as an exploratory and hypothesis-generating approach and the reformulation of the research questions make it more understandable and coherent. The implementation of the ‘limitations’ section helps the reader to understand the relevance of the results and how this study can serve as a preliminary approach to the research questions explored.

Validity of the findings

Approaching this study from a different point of view makes it now clearer how it can be used as a basis to broaden our knowledge of the use of pre-sleep music, sleep quality, psychological distress, and individual differences and behaviors. It is interesting to see how the topic can be explored further and broadened.

Additional comments

The new formulation of the study and of the research questions makes the exploratory approach of this study clearer. This gives the reader more insight into the methodologies used, the results, and the limitations, giving the study a more understandable and coherent context.

The authors addressed all the comments previously provided, improving both readability and comprehension of the paper.

---

## Round 0.3 · accepted · Accept

· Academic Editor

Accept

Thank you for carefully addressing all the raised points in your revision. I am pleased to confirm that your article is now accepted for publication. Congratulations on your excellent work!

Reviewer 3 ·

Basic reporting

Thank you for the clear revision of the manuscript. It now looks clear and has improved under every point of view.

Experimental design

Thanks for making the experimental design clear.

Validity of the findings

See previous review.